# Hotspot motion caused the Hawaiian-Emperor Bend and LLSVPs are not fixed

Richard K. Bono[1,2], John A. Tarduno [1,3] & Hans-Peter Bunge[4]

Controversy surrounds the fixity of both hotspots and large low shear velocity provinces (LLSVPs). Paleomagnetism, plate-circuit analyses, sediment facies, geodynamic modeling, and geochemistry suggest motion of the Hawaiian plume in Earth's mantle during formation of the Emperor seamounts. Herein, we report new paleomagnetic data from the Hawaiian chain (Midway Atoll) that indicate the Hawaiian plume arrived at its current latitude by 28 Ma. A dramatic decrease in distance between Hawaiian-Emperor and Louisville chain seamounts between 63 and 52 Ma confirms a high rate of southward Hawaiian hotspot drift (~47 mm yr$^{-1}$), and excludes true polar wander as a relevant factor. These findings further indicate that the Hawaiian-Emperor chain bend morphology was caused by hotspot motion, not plate motion. Rapid plume motion was likely produced by ridge-plume interaction and deeper influence of the Pacific LLSVP. When compared to plate circuit predictions, the Midway data suggest ~13 mm yr$^{-1}$ of African LLSVP motion since the Oligocene. LLSVP upwellings are not fixed, but also wander as they attract plumes and are shaped by deep mantle convection.

[1] Department of Earth and Environmental Sciences, University of Rochester, Rochester, NY 14627, USA. [2] Geomagnetism Laboratory, University of Liverpool, Liverpool L69 3GP, UK. [3] Department of Physics and Astronomy, University of Rochester, Rochester, NY 14627, USA. [4] Department of Earth and Environmental Sciences (Geophysics), Ludwig-Maximilians-University, Munich 80333, Germany. Correspondence and requests for materials should be addressed to R.K.B. (email: R.K.Bono@liverpool.ac.uk) or to J.A.T. (email: john.tarduno@rochester.edu)

Ocean drilling of the Emperor Seamounts provided the crucial paleomagnetic data defining motion of the Hawaiian hotspot in Earth's mantle[1]. Hotspot tracks on oceanic plates outside the Pacific Ocean basin had long been used to predict the trend of volcanic edifices emanating from the Hawaiian hotspot if it had been fixed[2]. These non-Pacific hot-spots must be tied to the Pacific plate via a plate circuit: a continuous linkage of plates separated only by spreading boundaries, with a history that can be reconstructed using marine magnetic anomalies. These analyses have yielded an enduring set of predicted locations that fail to duplicate the Emperor trend[3–7]. Plate-circuit analyses, however, can only detect motion of the Hawaiian hotspot relative to Indo-Atlantic hotspots. Paleomagnetic analyses of basalts recovered from ocean drilling[1,6,8–10], which are the gold standard for measuring paleolatitudes of oceanic plates, constrained this motion to the Hawaiian mantle plume (see Paleomagnetic analyses and the Pacific fixed hotspot reference frame, in Methods). The Emperor Seamounts lack coralline sediments above basement expected if they had formed at the present-day latitude of Hawaii[1]. Parés and Moore[11] concluded that eastern Pacific plate calcium carbonate accumulation was incompatible with a fixed Hawaiian hotspot. Geodynamic modeling of the entrainment and advection of a mantle plume conduit in mantle convection also suggested southward Hawaiian hotspot motion[5,12]. This class of model tends to underestimate the latitudinal motion relative to that recorded by the Emperor seamount paleomagnetic data, and requires a change in plate motion to create the ~60° Hawaiian-Emperor Bend (HEB)[1], points we return to below.

The consistency of these independent approaches indicating hotspot motion implies that absolute plate motion models based on fixed Pacific hotspots are problematic[13] (Methods). Still, efforts to preserve fixed hotspots and plate motion causation of the HEB continue. Gordon et al.[14] modeled marine magnetic anomaly skewness profiles and argued for a complex series of true polar wander (TPW) rotations and a fixed hotspot[15,16]. In contrast, Torsvik et al.[17] conclude that hotspot motion during creation of the Emperor Seamounts must be accepted, but they favor plume conduit entrainment and advection modeling results that yield a southward drift that is only ~25–60% of that documented by the paleomagnetic data. Torsvik et al.[17] argue that Emperor paleomagnetic data could be anomalous because of non-dipole fields, quoting results from Hawaiian lavas of the last ~425 kyr (HSDP2)[18], which show an apparent latitude anomaly of −3.6°. However, the uncertainty $\left(\begin{smallmatrix}+18.0°\\-12.9°\end{smallmatrix}\right)$, which is several times greater than the nominal anomaly, was not considered.

Herein, we first address the recording fidelity of the position of Hawaii by an analysis of cores recovered from drilling a site younger than the HEB bend: the ~28 million-year-old Midway Atoll. Data from Midway Atoll confirm the paleolatitude-recording accuracy of Hawaii and lend further support to the rapid episode of southward hotspot motion that formed the Emperor Seamount trend[1]. Next, we compare data from the Hawaii and Louisville chains[19–21] to evaluate and ultimately exclude TPW, confirming a hotspot motion causation for the HEB[6,22]. Finally, we return to the younger Hawaiian chain and show how our new results provide insight into the nature and fixity of large low shear velocity provinces[23] (LLSVPs). Specifically the new data support wander of the African LLSVP since the Oligocene. When combined with other global observations, these analyses indicate that LLSVPs are not fixed, but drift as they are affected by lower mantle flow.

## Results

**The paleolatitude of Midway Atoll**. Midway Atoll of the Hawaiian chain was drilled in 1965, with the principal result obtained at the Reef Hole (Fig. 1, Supplementary Fig. 1), where 120 m of lavas were penetrated[24] (see Drilling at Midway Atoll and prior analyses, in Methods). The recovered tholeiitic basalt flows show weathered flow boundaries; of special note is a thick soil horizon (minimum thickness of 5.8 m) between lava flows[25]. A shorter basalt section (~16 m) was recovered from the Sand Island Hole (Fig. 1, Supplementary Fig. 1). The highest quality K–Ar radiometric age data from mugearite and hawaiite from conglomerate pebbles above the Reef Hole basement yield an age of 27.6 ± 0.6 Ma[26,27]. This compares well with $^{40}Ar/^{39}Ar$ ages of 27.5 ± 1.2 Ma and 27.6 ± 0.9 Ma on shield phase samples recovered from subsequent dredging[28]. Grommé and Vine[29] sampled thirteen of the Midway lavas, applying only partial alternating field demagnetization on about one half of the samples. The nominal paleolatitude $\left(14.7^{+7.6°}_{-4.2°}\right)$ is generally not considered in recent analyses of Pacific plate motions because the paleomagnetic analysis techniques are incomplete (Methods).

We sampled throughout the sections of basalts, including weathered core tops and the thick soil (see Paleomagnetic sampling and analyses, in Methods). The magnetization directions show linear decay to the origin of orthogonal vector plots after the removal of spurious magnetizations at low unblocking temperatures (Fig. 2). Rapid intensity decay in the basalts commences over a range of temperatures (~275–400 °C) and extends to ~585 °C, indicating the dominance of titanomagnetite carriers. A small, and variable amount of remanence is carried by higher unblocking temperature minerals (indicative of hematite). Weathered flow tops, as well as soils, show a dominance of these higher blocking temperatures (>585 °C) highlighting hematite in these samples. The characteristic remanent magnetizations, determined by principal component analysis of the demagnetization data, record both normal and reversed magnetization zones (Supplementary Tables 1 and 2). Two positive contact tests (Fig. 2) and the different polarities strongly support to the preservation of a primary magnetization. Basalts from the Sand Island hole are of normal polarity and show demagnetization characteristics similar to those of the Reef Hole (Supplementary Tables 1 and 2, Methods).

We identify 12 lava flow means from the Reef Hole, and three from the Sand Island Hole (Supplementary Table 1). The 15 lava flow averages yield a mean paleoinclination of 30.1 ± 7.8° (Supplementary Table 3), corresponding to a paleolatitude of $16.2^{+5.1°}_{-4.6°}$ N, with an estimated angular dispersion of $S = 14.0^{+10.6°}_{-3.4°}$ (see Secular variation, in Methods). The mean $S$ value is close to, but slightly lower than predicted from data of the last 5–10 million years[30,31], much lower than predicted from Miocene-Oligocene data[32], and slightly lower than predicted from Oligocene-Eocene data[32] (Supplementary Figs. 2 and 3). Although there are numerous indicators that time has elapsed in the Midway basement stratigraphy, the $S$ estimates at best barely reach predicted time-averaged values. This reflects the limited number of temporally independent lava units (Supplementary Fig. 4) available rather than the age span of the basement penetrated.

Additional paleolatitude estimates, however, are available from the thick Reef Hole soil horizon. Magnetostratigraphy and geochronology suggest that the soil horizon represents weathering over at least several hundred thousand years during chron 9 (27.027–27.972 Ma)[33], a duration consistent with soil development on Hawaii[34] (see Duration of Reef Hole soil formation, in Methods). This age assignment suggests that the soil horizon represents time averaging that is equivalent to or greater than the

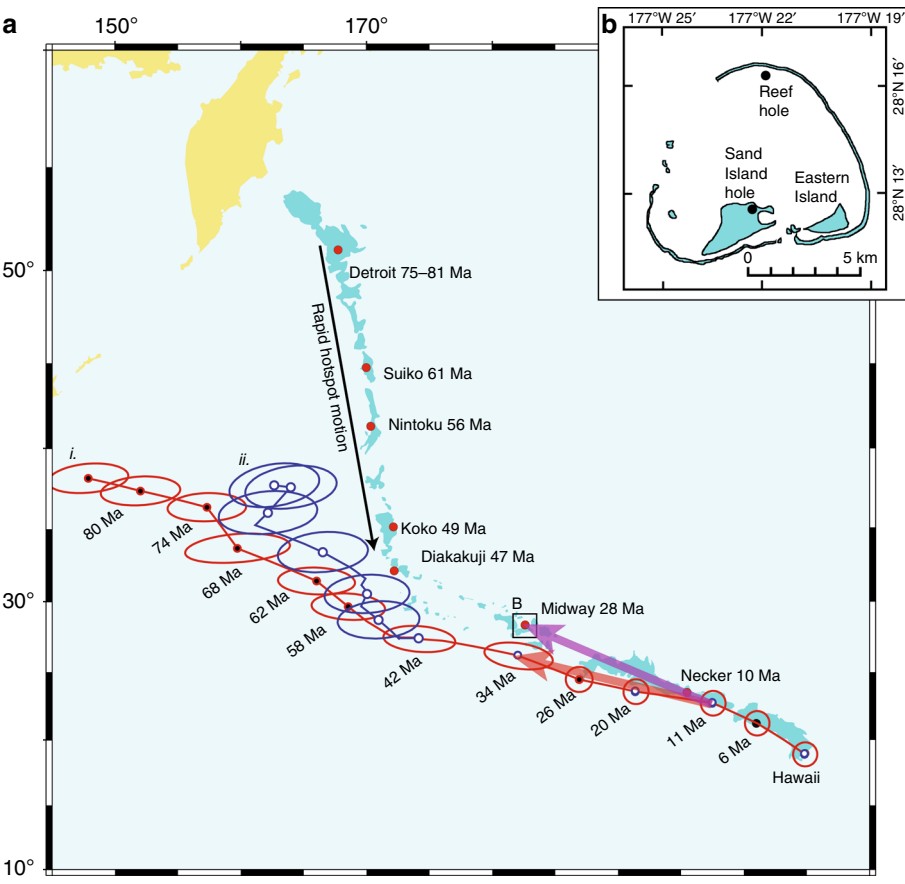

**Fig. 1** Location maps and plate-circuit predictions. **a** Hawaiian-Emperor chain, with episode of rapid hotspot motion highlighted. Predicted position of the Hawaiian hotspot using plate-circuit analyses (from ref. [6, 7, 13]); (i) East-West Antarctica[4]; (ii) Australia-Lord Howe Rise[5], using the updated Antarctica-Australia spreading history and rotations of Whittaker et al.[44] and Müller et al.[45]. Red and purple arrows mark differences in plate-circuit prediction and seamount trace. **b** Midway Atoll with Reef and Sand Island drill sites highlighted

cumulative sampling by the lava flows. In this case, it is most appropriate to give the soil horizon samples and lava means equal weight. This approach yields a paleolatitude of $19.2^{+3.7°}_{-3.4°}$ N (Supplementary Table 3), which is our preferred estimate. This value is indistinguishable from the present-day latitude of the Hawaiian hotspot (~19°). It also reveals as spurious the claim that the recording location of Hawaii is anomalous to the level it would affect paleomagnetic tests of hotspot drift[17]. It should also be noted that Cromwell et al.[31], in an analysis of volcanic data for the last 10 m.y. corrected for serial correlation effects (model LN3-SC), find a negligible anomaly at the location of Hawaii.

The Midway paleolatitude is very different from paleolatitudes recorded by drilling Emperor Seamount lavas (Fig. 3). Before moving on to an additional test, we first perform a formal analysis of the Emperor trend paleomagnetic data to determine a mean rate of latitudinal change. We use a Bayesian approach (specifically, Markov Chain Monte Carlo, see Methods)[35,36] to linear regression, incorporating uncertainties in age and paleolatitude. This yields a rate estimate of 47.8 (±15.3) mm yr$^{-1}$ (95% confidence). This analysis indicates that low rates of motion (i.e. 10 mm yr$^{-1}$) sometimes interpreted for hotspot drift during creation of the Emperor seamounts[14], have a negligible probability (less than 0.01%, Supplementary Fig. 5).

**Changes in inter-hotspot distance.** Paleomagnetic and radiometric age analyses of samples recovered during IODP Expedition 330 suggest only limited (3–5°) latitudinal drift of the

Louisville hotspot between 50 and 70 Ma[20]. This result highlighting the independent motion of the Hawaiian plume affords the possibility of measuring not just the relative motion documented by Konrad et al.[21], but also absolute motion. If the Louisville plume was only slowly moving while the Hawaiian plume was moving rapidly southward, this difference should be preserved as a change in distance between the volcanic edificies comprising the two hotspots tracks. Continued efforts to improve the age assignments of the Hawaiian-Emperor and Louisville Seamounts through $^{40}Ar/^{39}Ar$ radiometric analysis[20,21,28] allows this analysis (see Hawaii-Louisville seamount distances, in Methods). We select eight seamount pairs where the ages are within 3 m.y. (Supplementary Table 4). These data show a dramatic decrease in distance between 63 and 52 Ma (see Fig. 3, Supplementary Fig. 6, Methods) at a rate of 32.2 ± 6.7 mm yr$^{-1}$ (95% confidence interval). Distance differences between 80 and 70 Ma are indistinguishable and may point to a similar rate of southward hotspot motion of the two hotspots. There is a hint of this southward motion in the paleolatitude (42.9° S) from the oldest Louisville seamount drilled (Canopus)[20], which is more northerly than those of the younger seamounts, but the data do not fully average secular variation. Both tracks were probably influenced by ridges in the Late Cretaceous. After 25 Ma, distances between the hotspots increase, but at this time the Louisville hotspot magmatic output had waned to the point that lithospheric structure probably has a large effect on the plume location as recorded by its volcanic constructs.

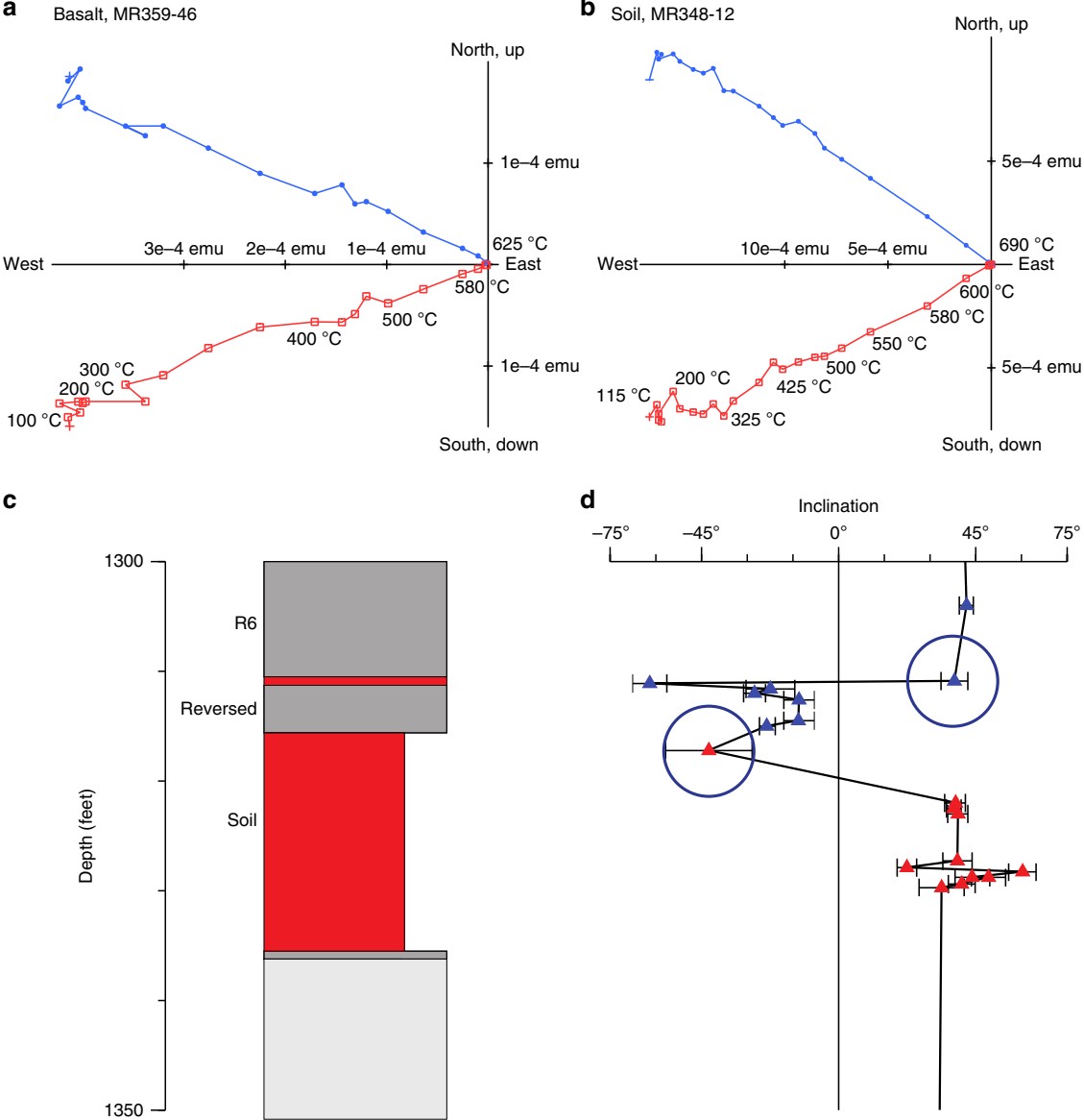

**Fig. 2** Paleomagnetic data and contact tests. Orthogonal vector plots of stepwise thermal demagnetization of basalt (**a**) and soil (**b**) samples. Red is inclination, blue is declination (cores are azimuthally unoriented). **c** Stratigraphy adjacent to thick soil horizon in the Reef Hole (Supplementary Fig. 1). Here the unit feet is retained as it was the unit of record. Red, soil or weather flow top. Dark grey, basalt. Light grey, unrecovered. **d** Characteristic remanent magnetization inclination. Blue, basalt samples. Red, soil samples. The uppermost soil sample from the thick soil horizon (circled) records the polarity of the overlying lava flow (reversed polarity) whereas, the lower samples record a different (normal polarity). This constitutes a positive contact test indicating that a primary magnetization has been preserved. Similarly, the uppermost sample of the weathered lava flow top of the reversed flow (circled) records the direction of the overlying lava flow (normal polarity), constituting a second positive contact test

**True polar wander revisited**. The change in hotspot distance[19,21] between 63 and 52 Ma provides spectacular confirmation of the rapid Hawaiian hotspot southward motion. Because this distance change is on a single plate it cannot reflect TPW, and therefore inferring that this process is responsible for the observed Emperor seamount paleomagnetic data trend[14] is incorrect. A principal set of analyses that have been used for continued calls for TPW have been marine magnetic anomaly skewness. Anomaly skewness values are model-dependent, as the data also reflect a complex mixture of oceanic crustal accretion processes. Nevertheless, the paleolatitudes predicted from the skewness-based models tend to scatter about the more robust paleolatitude trend defined by paleomagnetic analyses of seamount lava (Supplementary Fig. 7). However, these skewness models have also been used to call for a much younger TPW event (after 11 Ma) that would be responsible for North Hemisphere glaciation[37]. The new data from Midway also afford the opportunity to examine this hypothesized rotation. The closest skewness-based modeled paleolatitude to the age of Midway is a 32 Ma value (marine magnetic anomaly chron 12r)[38], which yields 22.3° N. Applying a Bayesian formalism, there is only an 8.9% chance of observing the skewness inclination given the distribution of paleomagnetic inclinations observed at Midway (Supplementary Fig. 8). Thus, we conclude that the apparent latitudinal discrepancy from the skewness models is inconsequential, and the related TPW rotation[14,37] unnecessary (see True polar wander based on marine magnetic anomaly skewness, in Methods).

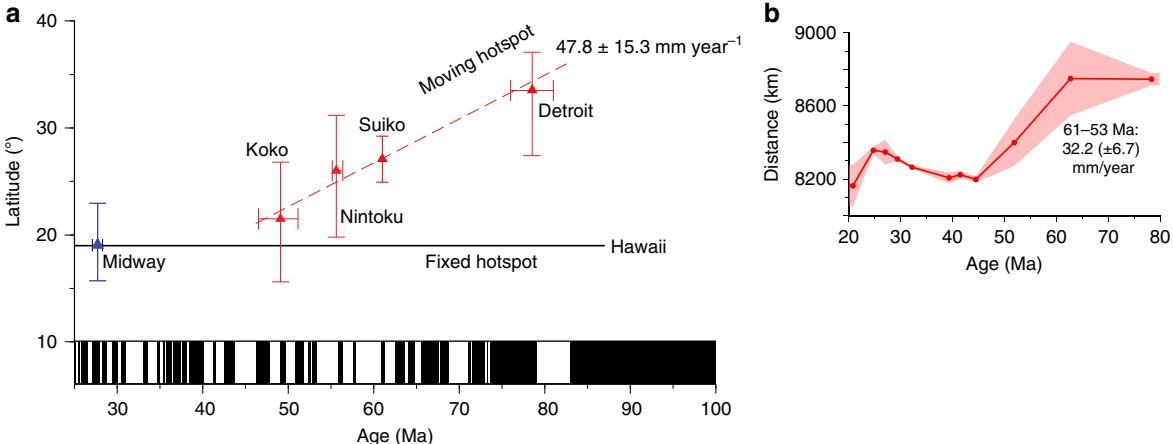

**Fig. 3** Paleomagnetic and inter-hotspot distance tests of hotspot fixity. **a** Paleolatitude versus time plot[1] with time-averaged results from paleomagnetic analyses of samples recovered from scientific drilling of seamounts augmented with paleolatitude value from Midway Atoll (blue triangle) presented here. Age uncertainty shown is 2 sigma, paleolatitude uncertainty is 95% confidence. **b** Distance between Hawaiian-Emperor and Louisville seamounts with formation ages within 3 Myr with 95% confidence. (Supplementary Table 4)

**Creating the Hawaiian-Emperor Bend morphology**. The confirmation of rapid hotspot motion indicates that this process must be central to the formation of the HEB. This conclusion is in concert with the lack of clear geological evidence for a profound event, on the Pacific margins[39] or in the seafloor spreading record[40], corresponding to the age of the HEB and commensurate with its prior designation as the largest and most rapid change in plate motion for a large oceanic plate. Some of the confusion about the HEB arises from a narrow focus on relative plate motions[41] or ages[42] immediately before and after the bend. As a result, the HEB is considered out of context with respect to the overall morphology of the Hawaiian-Emperor chain.

Plate-circuit predictions provide the means of understanding the origin of the overall morphology. They can be used to illustrate a hypothetical Hawaiian-Emperor chain produced by a hotspot fixed in the deep mantle. The standard plate circuit through East Antarctica shows that the HEB (Supplementary Fig. 9A) is replaced by a small wiggle in the chain[6] (Supplementary Fig. 9B) similar to those before and after 47 Ma. Thus, even if there is a change in absolute plate motion direction at ~47 Ma, the change appears to have been imprinted in the volcanic track as only a wiggle similar to those that can be related to early and late Paleogene plate motion changes[43]. Similar conclusions can be reached based on the alternative plate circuit, through Lord Howe Rise[5], updated by the revised Antarctic-Australian spreading history[44,45] (Supplementary Fig. 9C).

While plate motion changes played only a minor role in the HEB morphology, the analyses here do not exclude the presence of plate motion changes in the Pacific basin. Detailed analyses of Indo-Atlantic hotspot tracks fail to find a signal, which might indicate an important global change in absolute plate motion corresponding to chron 21 (47.9 Ma) and the HEB[46]. But the still nascent Pacific apparent polar wander path does hint at an acceleration of the Pacific plate in the Cenozoic. Cottrell and Tarduno[47] noted that very slow Pacific apparent polar wander in the Late Cretaceous was followed by faster early Cenozoic motion. This is largely consistent with the development of subduction zones in the western Pacific[48].

**Causes of the rapid hotspot motion**. There are are least two processes that can explain the rapid hotspot motion, and its sudden slowdown, at the HEB. They are not mutually exclusive.

In the first the ancestral Hawaiian plume is drawn at mid-mantle depths toward rapid spreading at the Late Cretaceous Kula/Pacific ridge[6], a process supported by numerical simulations[49]. As spreading wanes, and eventually ceases altogether, the ridge influence on the plume diminishes. During this time of ever diminishing influence, the plume moves toward a more vertical position in the mantle. In the second, the plume position is affected by motions in the deeper mantle[6]. Numerical modeling of Hassan et al.[50] shows that the interaction of the plume with the Pacific LLSVP can give rise to the observed motions. Geochemistry lends support to this interpretation. The Hawaiian Islands are characterized by two distinctive lava trends, enriched Loa lavas and less primitive Kea lavas, best characterized by $^{208}$Pb*/$^{206}$Pb*, which gauges radiogenic ingrowth in the formation of Earth[51]. The Emperor trend seamounts lack the Loa trend signature. Although there are several ways to explain double geochemical traces on young volcanic chains[52], Harrison et al.[53] have focused on a mechanism that can give rise to the observed longterm temporal signature without a change in plate motion. Southward motion and anchoring of the plume on the LLSVP (Fig. 4a) results in the gradual entrainment of LLSVP material and the appearance of Loa lavas on the Hawaiian chain[53]. This explanation is viable if the Loa geochemical anomaly is a passive marker[54] of subducted material stored in the Pacific LLSVP. The potential for LLVSP deformation implicit in the Hassan et al.[50] model raises the question of whether they are fixed in the deep mantle, a question that we address below.

## Discussion

To understand the implications of our new data from Midway Atoll, we recall the predictions of the Hawaiian-Emperor chain positions using plate circuits and the Indo-Atlantic hotspots (Fig. 1). Because the Midway paleomagnetic data indicate the hotspot had arrived at its present-day latitude by 28 Ma, there should be no discrepancy between the predicted and actual positions of the the Hawaiian-chain volcanoes. However, a systematic discrepancy is clear (red vs. purple arrows, Fig. 1). Using the Bayesian formalism (Supplementary Fig. 10), we determine that the probability that the offset is compatible with the paleomagnetic data is small (8.4%). If the discrepancy between actual and predicted locations of Hawaiian volcanoes younger than 28 Ma is not due to drift of the Hawaiian hotspot, the alternative explanation is motion of the Indo-Atlantic hotspot group, rooted

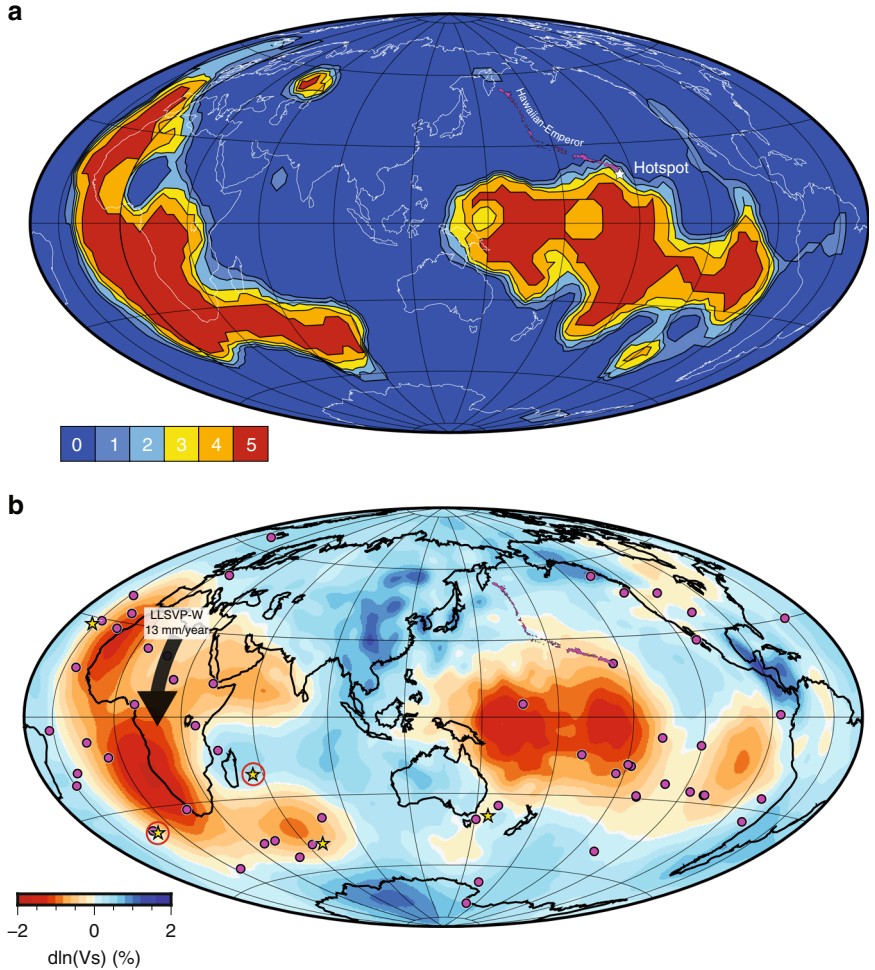

**Fig. 4** Large low shear velocity province wander. **a** Voting map cluster analysis of large low shear velocity province (LLSVP) morphology[79] with Hawaiian-Emperor chain highlighted, sampled at 3 degree intervals. Map shows cluster analysis of five shear velocity ($V_s$) tomographic models evaluated at 2700 km depth. **b** SEISGLOB2 $V_s$ tomographic model of Durand et al.[65] resolved at 2700 km depth, with hotspots (purple circles). Yellow stars show those hotspots used for the Indo-Atlantic reference frame of Müller et al.[80]. Red circles are hotspots used in Indo-Atlantic reference frame of Steinberger[81]

in the African LLSVP. This LLSVP Wander (LLSVP-W) (Fig. 4b) occurs at a minimum (latitudinal) rate of ~$12.9 \pm 7.5$ mm yr$^{-1}$ (Supplementary Fig. 11).

The latitudinal motion corresponds to a distance of some 1300 kilometers at Earth's surface when averaged over 100 million years, and thus LLSVP-W is important when seeking to backtrack the position of continents. This supports the conclusion of Smirnov and Tarduno[55] who performed a paleomagnetic Euler pole backtracking to ~250 Ma, and found large differences in longitude (~40°) between predictions and those derived from a fixed Africa/LLSVP, and motion of Atlantic mantle plumes at rates of $34 \pm 9$ mm yr$^{-1}$ during the mid-Cretaceous has been observed from an analysis of paleomagnetic data[56]; this motion is also predicted from numerical simulations[57]. The paleomagnetically constrained velocities are compatible with retrodictions of lower mantle flow[58]. It is important to note that the large scale (>1000 km) of coherent hotspot motion[56] implies a wholesale motion of LLSVPs[59], rather than deformation by lower mantle flow alone. Thus, while long-lived (>100 m.y.), LLSVPs should not be thought of as constant fixed features. Instead, they can exhibit a net slow motion punctuated by episodes of rapid wander.

The rates, pattern, and scale of these motions have implications for the nature and origin of LLSVPs. Although their base (~100 km)

may be chemically distinct, seismic evidence[60–62] is consistent with LLSVPs being dominated by thermal heterogeneity. Global geodynamic flow models predict that the associated upwelling flow induces horizontal motion in the deep mantle directed towards the provinces[63]. This deep mantle lateral motion helps to explain the migration of plumes reported in high Rayleigh number 3-D spherical mantle convection models[64]. LLSVP-W indicates that any marked chemical difference at the base of LLSVPs is insufficient for fixity; slip at the core-mantle boundary is possible.

The lower mantle LLSVP structure is more complex[65] than that of spherical harmonic degree 2 (Fig. 4b), with subregions of lower shear velocity. Clusters of upwelling plumes are also seen in compressible global mantle circulation models with purely thermal piles[66]. Moreover, mantle flow models over the last 230 m. y.[59] reproduce the modern LLSVP structure, signaling that post-Pangea subduction has focused lower mantle upwellings beneath Africa and the Pacific. This result emphasizes that it is dynamically infeasible to have fixed LLSVPs and evolving plate configurations at the same time[67]. Much of the hotspot distribution seen today may be the end result of plume motion toward LLSVP subregions of focused upwelling, with the Hawaiian plume being our best example of motion from a distant location (i.e. ~1500 km) to a LLSVP.

## Methods

**Paleomagnetic analyses and the Pacific fixed hotspot reference frame.** Kono[8] conducted a detailed paleomagnetic study of basalts recovered from drilling of Suiko Seamount and obtained a paleolatitude of ~27° N, but he made no interpretation of this value with respect to hotspot fixity. Tarduno and Cottrell[9] obtained a paleolatitude of ~33° N for basalts recovered from Detroit Seamount of the northern Emperor Seamounts, and, together with the value from Suiko Seamount, suggested that the trend of the Emperor seamounts largely reflected motion of the Hawaiian plume in Earth's mantle. This interpretation was subsequently supported by age and paleomagnetic analyses of basalts recovered by drilling of the Emperor trend during Ocean Drilling Project (ODP) Leg 197[1,6,10]. Doubrovine and Tarduno[13] showed that one commonly used Pacific plate motion model, WK08, does not accurately predict the location of the Hawaiian hotspot for times older than the time of the Hawaiian-Emperor bend, which implies inter-hotspot motion. This issue was acknowledged by the model authors, who discussed relative motion between the Louisville and Hawaiian hotspots[19].

**Drilling at Midway Atoll and prior analyses.** Midway Atoll, located nearly equidistant from North America and Asia (Fig. 1), was explored by drilling through the carbonate reef into underlying basalt flows[24,25]. At the Reef Hole, located in the north of the Midway lagoon, drilling was conducted from a steel barge flooded with seawater, penetrating 1654 feet (504 m), of which the bottom 393 feet (120 m) was basalt. The unit feet was the measurement of record. The Sand Island Hole was positioned on the larger island forming the atoll (Fig. 1), where a thinner sediment and basalt section (568 feet; 173 m) was recovered. The sedimentary section above the basalt flows represents a classic subsidence succession: conglomerates with basalt clasts give way to volcanogenic sediments and finally to Miocene-Pleistocene limestone and dolomite.

Grommé and Vine[29] combined results from partially demagnetized samples with undemagnetized natural remanent magnetization (NRM) directions to obtain a mean direction. No thermal demagnetization results were reported. The thick soil horizon was not sampled, nor were the highly weathered lava flow tops. The use of NRM directions is not accepted in modern analyses because these can be contaminated by spurious overprints, occluding the primary magnetic signature. We were unable to locate the samples assigned to reversed polarity by Grommé and Vine[29] in the Sand Island Hole.

**Paleomagnetic sampling and analyses.** Midway cores were stored at the University of Hawaii in their original wooden boxes; for most of the sections examined, it appears that the cores had remained untouched since drilling and initial sampling. A few sections, however, could not be located. Although large continuous sections of rock were recovered, all material we examined was heavily weathered, hindering the collection of drilled cores. Instead, 1-cm cubes were prepared with a rock saw using a bronze-blade impregnated with diamonds at the University of Rochester for analysis. All samples were measured using a 2G 3-component DC SQUID magnetometer with high resolution coils in the magnetically shielded room at the University of Rochester (ambient field <200 nT). Given the weathered nature of the cores, we focused on thermal demagnetization in air using ASC Thermal Demagnetization ovens.

**Secular variation.** Having demonstrated, both through the presence of reversed and normal polarity lava flows, and contact tests, that a primary magnetization is preserved in the Midway Atoll cores, an important next step is to assess how much time is recorded. This is essential because of the inherent variation of the field on short time scales. Specifically, the geomagnetic field at a radius $r$, colatitude $\theta$, longitude $\phi$, and time $t$ can be described by the gradient of the scalar potential ($\Phi$):

$$\Phi(r, \theta, \phi, t) = r_e \sum_{l=1}^{\infty} \sum_{m=0}^{l} \left(\frac{r_e}{r}\right)^{l+1} P_l^m(\cos\theta)[g_l^m(t)\cos m\phi + h_l^m(t)\sin m\phi] \quad (1)$$

where $P_l^m$ are partially normalized Schmidt functions, $r_e$ is the radius of Earth and the Gauss coefficients $g_l^m$ and $h_l^m$ describe the size of spatially varying fields. Thus, the record of any one lava flow could depart widely from the value dominated the axial dipole (i.e. $g_1^0$) needed to accurately gauge paleolatitude. The angular dispersion $S$ of virtual geomagnetic poles (VGPs) is typically used to evaluate how results from a set of lavas represents the past magnetic field:

$$S = \sqrt{\frac{1}{N-1}\Sigma\Delta_i^2} \quad (2)$$

where $N$ is the number of individual VGP's and $\Delta_i$ is the angle between the $i$th VGP and the mean paleomagnetic pole. If a given lava set represents a brief interval of time, it will tend to have an $S$ less than global reference values known to average the geomagnetic field (once an average has been obtained, increasing the number of observations increases the precision of the mean, but $S$ remains the same).

We combine two of the Reef Hole lavas (R10 and R9) into a single mean because they have nearly identical directions and there is no geological indicator that significant time elapsed between their eruption. From the basalt flow paleomagnetic results, a synthetic dataset ($n = 14$) was drawn with replacement. For each draw, the mean inclination, precision parameter, k (following the

inclination-only method of McFadden and Reid[68]), and VGP dispersion (S) were calculated following the methods of Tarduno and Sager[69]. This process was repeated 1000 times[70] to yield uncertainties. To assess the degree in which individual basalt flows contributed to VGP dispersion (S), the relative percent deviation from a uniform distribution was determined for each flow following the bootstrap resampling. The results of the bootstrap were divided into 1 degree bins based on the calculated S (Supplementary Fig. 4). The total number of sample draws for each bin was then determined, along with the percentage for each flow in each bin. The expected percentage for each flow assuming a uniform distribution was then subtracted from the percent observed, yielding a relative deviation from uniform distribution for each flow for each S bin. Flows which contribute more frequently to a given S value will yield a positive deviance (and conversely, a negative percentage means that a flow is less commonly found in draws which yield that S value).

**Duration of Reef Hole soil formation.** The radiometric age data are most compatible with the construction of the Midway ediface during chron 9n (27.027–27.972 Ma)[33,71]. The simplest interpretation of the stratigraphy and magnetic record is that the older basalts of the Reef Hole were erupted in the early part of chron 9n, and that the soil represents an interval of weathering hundreds-of-thousands-of-years long. This weathering extended into chron 8r, marked by the reversed Reef Hole lava, and was followed by another hiatus in lava emplacement, followed by the eruption of the uppermost Reef Hole lavas during chron 8n.2. In an analysis of six modern sites on the Hawaiian Islands at 1200 m elevation, soil exceeding 5 m in thickness were reported only from the oldest (4.1 Ma) site[34]. While several factors could have resulted in a faster soil accumulation at the Midway site ~28 million years ago, these modern observations suggest the thick Reef Hole soil could have formed over enough time to record a significant amount of secular variation.

**Markov Chain Monte Carlo (MCMC).** Here, a model is specified following the form of a linear regression along with a series of priors, which represent a description of the assumed distributions for each model parameter. A set of model parameters are randomly drawn from the prior distributions, and the likelihood of the model realization is determined. Model realizations are retained based on an acceptance parameter, generating a Markov Chain. The resulting accepted model parameters comprise a posterior probability distribution with realistic uncertainty estimates. Paleolatitudes (and their corresponding uncertainties) are treated as observations and seamount ages as independent variables; age uncertainties (from radiometric data) are assumed to be drawn from a normal distribution. Rate of motion, uncertainty, and intercept are treated as model parameters (with broad, zero-mean normal priors). The likelihood estimate is assumed to be described by a normal distribution. In the case of modeling LLSVP motion, the predicted latitude is determined by finding the latitudinal difference between the predicted seamount latitude using the plate-circuit reconstruction[13] and the paleomagnetic paleolatitude for the Hawaiian hotspot after the HEB. Sampling was performed using the gradient-based No-U-Turn-Sampler using the PyMC3 framework[72] generating 10,000 samples, with only the final 25% of the samples retained to accommodate burn-in and mixing of the MCMC.

**Hawaii-Louisville seamount distances.** As aforementioned, Doubrovine and, Tarduno[13] showed the inconsistency of the Hawaii hotspot for times older than the HEB in fixed Pacific hotspot models, implying inter-hotspot motion. This motion was quantified by Wessel and Kroenke[19] who examined distances between the Louisville and Hawaiian hotspots. Konrad et al.[21] performed an analysis of inter-hotspot distance between the Louisville and Hawaiian-Emperor seamounts, with two key differences from our approach: best-fit, synthetic age progressions were developed for each chain from which the resulting great circle distance was calculated for coeval ages, and the interval of analysis spanned 60–48 Ma. We feel our approach is more conservative in that we do not use synthetic age progressions. Notwithstanding the differences in approach, Konrad et al.[21] yield a rate estimate with a 1-sigma uncertainty (53 ± 21 mm yr⁻¹) that includes the rate we have determined. Finally, we note that some models infer an eastward component of Louisville hotspot drift[73]. In our approach we explicitly attempt to separate observations from models. However, if the Louisville hotspot drifted eastward between 63 and 52 Ma, our calculations underestimate the rates of convergence and Hawaiian hotspot motion.

**True polar wander based on marine magnetic anomaly skewness.** Various TPW scenarios have been posed based on analyses of marine magnetic anomalies, the most recent[74] being two rotations, an older one, after the formation of the Emperor seamounts, and a much younger event, after 11 Ma[37]. As documented in the main text, the changes in seamount distance exclude the older proposed TPW as being an important process. To examine the younger rotation, we select the marine magnetic anomaly skewness pole closest in age to Midway. The skewness-based paleolatitude prediction is north of the nearly coeval value from Midway Atoll, and the reported model paleolatitude uncertainty is extraordinarily small (approximately $^{+1.4°}_{-1.5°}$). However, it must be kept in mind this is a model uncertainty rather than an uncertainty in paleolatitude from paleomagnetic analysis, which has

physical meaning (i.e. geomagnetic secular variation). The small probability of the paleolatitude predicted from the skewness model based on data from Midway Atoll indicates that the reported skewness uncertainties do not capture the true range of paleolatitude error. These conclusions are consistent with the long-held inference that the uncertainties in skewness-based models are much greater than reported parameters[75]. Overall, our new analysis of the Midway Atoll, and inferences from the Hawaiian-Emperor chain, support prior conclusions of only minor TPW since the mid-Cretaceous[56,76], consistent with mantle circulation models that highlight the dampening of TPW[77,78].

## Data availability

Data presented here are available in the Earthref (MagIC) database (earthref.org/MagIC/16656). Source data underlying figures are also provided as a Source Data File.

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

## Acknowledgements

We thank David Clague for alerting us that the cores from the Midway Atoll drilling project still existed, Sandy Shor and the staff at the University of Hawaii for assistance in sampling, and Siavash Ghelichkhan and Bernhard Schuberth for insightful discussions. Support for this work has been partially funded by GSA Harold T. Stearns Fellowship to R.K.B. and NSF EAR 1656348 to J.A.T.

## Author contributions

J.A.T. conceived and supervised the project. R.K.B. and J.A.T. sampled the Midway cores. R.K.B. conducted rock magnetic and paleomagnetic measurements; data were analyzed by R.K.B. and J.A.T. H.-P.B. contributed to the geodynamic interpretations. All authors contributed to the writing of the manuscript.

## Additional information

**Competing interests:** The authors declare no competing interests.

