## [Peer Review File · Nature Communications]

Reviewers' comments:

Reviewer #1 (Remarks to the Author):

This paper by Bono et al. provides an excellent summary and a clean opinion on three key science questions: (1) does Hawaiian hotspot motion (and not a change in absolute plate motion) explain the Hawaiian-Emperor Bend (HEB), (2) is true polar wander (i.e. a sudden tilt of the entire Earth) a process that has occurred during the last 70 Myr, and (3) are LLSVPs long-lived, fixed, deep mantle features? To do so the authors focus on two of the most direct pieces of evidence that we can lay our hands on to help us answer these questions, namely (1) paleolatitude estimates based on paleomagnetic inclination data from series of lava flows drilled with scientific ocean drilling and (2) inter-hotspot distance measurements between two hotspot trails that have been coeval during their histories and are residing on the same rigid tectonic plate. Through this approach they show strongly that hotspot motion of Hawaii alone is the likely process that caused the HEB, they show that since the plume motion compared between Hawaii and Louisville are so different that any paleolatitude signal cannot be caused by TPW, consistent with the findings of Koppers et al. (2012; see discussion in their supplement), and they show that the LLSVPs themselves are likely not fixed on time scales of 100 Myr. The addition of the Midway paleolatitude data is an extremely valuable data set that clearly limits the plume motion to the older end of the Hawaiian-Emperor seamount trail and that is consistent with the inter-hotspot-distance data sets (e.g. Konrad et al. 2018). All data sets used in the opinion of these authors are indeed model independent, which is the key strength here, and thus provide the cleanest possible kinds of observation. It is time for the community to adopt the moving hotspot concept and focus their minds on mapping out plume and LLSVP behavior over geological time. This paper lays the groundwork for a fruitful future research avenue in geodynamics and our understanding of the Earth's deep dynamics, convection, and make up. This is an excellent article, definitely provocative, and perfect for Nature Communications. It is very well written and I recommend publication with very minor revisions. The only minor disappointment, as a geochronologist, is that the authors have not $^{40}\text{Ar}/^{39}\text{Ar}$ age dated the Midway cores, to add a more definite time stamp on this key observation!

Very minor comments on the text and some comments have been included in an annotated PDF

Anthony A.P. Koppers.

Reviewer #2 (Remarks to the Author):

Review of Bono et al. (2019) – Hotspot motion caused the Hawaiian-Emperor Bend and LLSVPs are not fixed

This manuscript provides new paleomagnetic data for an ancient Hawaiian plume derived seamount. The manuscript speaks to the debate over the origin of the Hawaiian-Emperor Bend (HEB) as well as the role of large low shear-wave velocity provinces play in modulating plume motion. The manuscript is well written with a clear and concise story. This manuscript is suitable for publication in Nature Communication with some minor revisions detailed below.

Major Points

Using changes in inter-hotspot distance for absolute plume motion:

If you wish to state absolute plume motion derived from comparing the geometry of Louisville to Hawaii, the authors need to clearly state that there is no eastward drift of the Louisville hotspot assumed. This eastward drift is predicted by Steinberger (2000) plume motion models as well as a slightly eastward, but mostly southward, drift from Hassan et al. (2016). If you model some eastward drift affecting the Louisville plume you will get slightly different values, although it is likely within uncertainty of your estimates.

Chemistry of the Emperor Chain Seamounts:

There are alternative views to explain the chemical composition of the Emperor Seamounts that do not invoke changes plume location relative to the LLSVP. Regelous et al. (2003) interpreted the chemical systematics of the Emperor Chain seamounts as resulting from changes in the thickness of the lithosphere the Hawaiian plume was interacting with. As the hotspot moved further way from the thinner lithosphere formed from the ancient Kula/Pacific Ridge, the degree of melting decreased resulting in more heterogeneous enriched magmas similar to what is observed on younger Hawaiian volcanos. This interpretation of the chemistry more closely supports a ridge capture/release scenario, although there is no chemical evidence that the plume was on the ridge prior to ~60 Ma.

Work by Jones et al. (2017) argued that the Kea-type magmas (similar to the Emperor Chain) are formed from a difference in melting regime of the plume. To briefly summarize, the kea type magmas originate from hotter, deeper peridotite while the Loa lava flows form from shallower more-enriched pyroxenites. The geochemical trends result from changes in plate motion relative to the shearing of the mantle plume. Neither of these scenarios invoke changes in deep mantle dynamics.

Plume attraction to LLSVPs:

I advise caution with attributing plume motion to plume migration towards an LLSVP edge. I have not seen any models that predict this phenomenon (although I may have missed some). Harrison et al. (2017) suggests that method but doesn't provide a geodynamic simulation to support it. Furthermore, Harrison et al. (2017) suggest the LLSVPs contain recycled subducted material, which is currently heavily debated (e.g. Li and McNamara, 2013; Davies et al. 2015; Rizo et al., 2016). However, many geodynamic models predict the initiation and anchoring of plumes on LLSVP margins (e.g. Steinberger et al. 2012; Tan et al. 2011). Therefore, I feel either more references are needed or more information supporting the claims in the abstract and text that LLSVP's "...wander as they continually attract plumes...".

Line by line points:

Abstract: missing 'shear' in 'large low velocity province (LLSVP)'

Figure S6: What is the hotspot location for Louisville? The current hotspot location is debated, assuming you're using the data from Konrad et al. (2018) it's based on the location suggested in Koppers et al. (2011).

Pedantic/Minor points:

None, paper is well written and edited.

Thanks for the interesting read,

Kevin Konrad

Reviewer #3 (Remarks to the Author):

This is an excellent paper which was a joy to read. It presents important new data to clarify the cause of the Hawaii-Emperor Bend (HEB), a major ongoing controversy. The paper is clearly written and well-illustrated, and follows clear, deductive scientific reasoning, unlike some other papers that have been published on this subject. This paper forms an important milestone to cement our understanding of the HEB origin. I recommend that this paper be published essentially as is, with some very minor corrections.

Under the heading “Creating the HEB Morphology” I believe that an incorrect reference is given:

The confirmation of rapid hotspot motion indicates that this process must be central to the formation of the HEB. This conclusion is in concert with the lack of clear geological evidence for a profound event, on the Pacific margins³⁹ or in the seafloor spreading record⁴⁰,

Ref 40 is the reply to a comment on Wright et al.’s *Geology* paper from 2016 (ref 43). Ref 43 should be cited here instead of ref 40.

In the same section, there is this statement:

Similar conclusions can be reached based on the alternative plate circuit, through Lord Howe Rise⁵, updated by the revised Antarctic-Australian spreading history⁴⁴ (fig. S9C).

Even though I agree with the general statement, I should warn the authors that they should not be using the rotation from ref 44 (Whittaker et al., *Science*, 2007). Unfortunately there were some oversights in this paper, and the rotations we derived at the time did not work entirely well with fitting the westernmost portion of the SEIND spreading system between the Kerguelen Plateau and Broken Ridge. Therefore we later revised these rotations and designed them such that the Kerguelen Plateau-Broken Ridge region would be fitted better. The reference is:

Whittaker, J.M., Williams, S.E. and Müller, R.D., 2013, Revised Tectonic Evolution of the Eastern Indian Ocean, *Geochemistry, Geophysics, Geosystems*, 14, 6, 1891-1909.

These rotations are also used in our current global plate model

Müller, R.D, Seton, M., Zahirovic, S., Williams, S.E., Matthews, K.J., Wright, N.M., Shephard, G.E., Maloney, K.Y., Barnett-Moore, N., Hosseinpour, M., Bower, D.J. and Cannon, J., 2016, Ocean basin evolution and global-scale plate reorganization events since Pangea breakup, *Annual Review of Earth and Planetary Science*, Vol 44, 107-138. DOI: [10.1146/annurev-earth-060115-012211](https://doi.org/10.1146/annurev-earth-060115-012211).

And the associated rotation file be downloaded here:

<https://www.earthbyte.org/category/resources/data-models/global-regional-plate-motion-models/>

under “Müller et al. (2016) Late Triassic to present-day global model (Annual Review of Earth and Planetary Sciences)”

I recommend that the authors redo their associated calculations with these rotations, as otherwise any lines of arguments that depend on these rotations could easily be shot down because the 2007 rotations are known to be flawed (even though the basic arguments in the paper still hold) and slight changes in these rotations can have very non-linear effects on far-away places on the Pacific Plate.

In the second-last paragraph if the paper:

LLSVPs should not be thought of as constant fixed features. Instead, they can exhibit a net slow motion punctuated by episodes of rapid wander

I would recommend to modify this statement in the following sense. One of the points of the Hassan et al. (2016) paper was to show that rather than wholesale motion of LLSVPs, they appear to undergo deformation, and their sides or flanks that are most proximal to active subduction zones are much more prone to deformation and migration than sides far away from active subduction. Wholesale motion of LLSVPs might have occurred as well in addition, and we see this most clearly with the example of the Perm anomaly, a small LLSVP which formed in response to a network of subduction zones which completely surrounded it until the Mongol-Okhotsk Ocean closed. The Perm anomaly is not in the place now where reconstructions (actually anyone’s reconstruction) place the girdle of subduction zones that likely caused it, so it must have experienced wholesale lateral motion (Flament et al., 2017). This is an inescapable conclusion, unless one wanted to argue that the Perm anomaly has nothing to do at all with Mesozoic subduction history, and just happens to be in a nearby place to this now extinct subduction girdle by pure coincidence. So there is evidence for both (LLSVP asymmetric deformation and wholesale motion) and both processes may cause hotspot wander. It might be worthwhile making this point.

Last paragraph:

A long history of subduction has likely focused lower mantle upwellings beneath Africa and the Pacific.

I find this sentence ambiguous. What is a “long history”? Is this meant to be 100 or 500 million years? Even though I know what the authors mean to imply, it may not be so clear to a general readership. I think this argument should be made much more strongly and clearly. Many published mantle convection models run over ~200 my produce the current overall LLSVP structure. Amongst all the models we have run in our group over more than a decade (easily more than 100, and many are published) one would not find a single one that did not result in this outcome. So one has to ask oneself: 200 million years of post-Pangea convection provides the overall geometry of the LLSVPs as they are observed, and, as recently pointed out in a paper on Stoneley wave splitting by Koelemeijer et al. (Nature Comms, 2017), they likely have an overall density that is lower than the surrounding mantle, not higher, as often assumed. This is in line with convection models that produce LLSVPs primarily as thermal upwellings, even though their base is likely chemically distinct – see recent EOS piece on this topic. So, simple deductive reasoning then tells us that if subduction history has such a strong influence on shaping LLSVPs, indeed creating them as they exist today, then obviously they cannot be fixed through time, as subduction geometry is not fixed in time. The point should to be made more strongly towards the end of the paper.

Response to Reviewer Comments

Reviewer 1 (Anthony Koppers)

Comment: “This paper by Bono et al. provides an excellent summary and a clean opinion on three key science questions: (1) does Hawaiian hotspot motion (and not a change in absolute plate motion) explain the Hawaiian-Emperor Bend (HEB), (2) is true polar wander (i.e. a sudden tilt of the entire Earth) a process that has occurred during the last 70 Myr, and (3) are LLSVPs long-lived, fixed, deep mantle features? To do so the authors focus on two of the most direct pieces of evidence that we can lay our hands on to help us answer these questions, namely (1) paleolatitude estimates based on paleomagnetic inclination data from series of lava flows drilled with scientific ocean drilling and (2) inter-hotspot distance measurements between two hotspot trails that have been coeval during their histories and are residing on the same rigid tectonic plate. Through this approach they show strongly that hotspot motion of Hawaii alone is the likely process that caused the HEB, they show that since the plume motion compared between Hawaii and Louisville are so different that any paleolatitude signal cannot be caused by TPW, consistent with the findings of Koppers et al. (2012; see discussion in their supplement), and they show that the LLSVPs themselves are likely not fixed on time scales of 100 Myr. The addition of the Midway paleolatitude data is an extremely valuable data set that clearly limits the plume motion to the older end of the Hawaiian-Emperor seamount trail and that is consistent with the inter-hotspot-distance data sets (e.g. Konrad et al. 2018). All data sets used in the opinion of these authors are indeed model independent, which is the key strength here, and thus provide the cleanest possible kinds of observation. It is time for the community to adopt the moving hotspot concept and focus their minds on mapping out plume and LLSVP behavior over geological time. This paper lays the groundwork for a fruitful future research avenue in geodynamics and our understanding of the Earth’s deep dynamics, convection, and make up. This is an excellent article, definitely provocative, and perfect for Nature Communications. It is very well written and I recommend publication with very minor revisions.

Response: We thank the reviewer for this assessment.

Comment: “The only minor disappointment, as a geochronologist, is that the authors have not $^{40}\text{Ar}/^{39}\text{Ar}$ age dated the Midway cores, to add a more definite time stamp on this key observation!”

Response: We agree that additional ages are desirable. However, we reiterate the remarkable agreement in age between the highest quality Kr-Ar results (Dalrymple et al., 1977; Clague and Dalrymple, 1989) from the Midway Reef Hole and the $^{40}\text{Ar}/^{39}\text{Ar}$ age data from O’Connor et al. (2013) on Midway dredged basalts, suggesting adequate age control to support our conclusions. We note that Midway lavas recovered by drilling are weathered. While this is favorable for soil formation and averaging of time for the paleomagnetic analysis, it makes geochronology challenging as target minerals (e.g., plagioclase feldspars) are altered. Nevertheless, some volcanic clasts in the conglomerate from the Reef Hole yielded the Kr-Ar data quoted and might be explored in future studies.

Comment: “Very minor comments on the text and some comments have been included in an annotated PDF”

Response: We thank the reviewer for these minor text comments which have been considered in revising the manuscript.

Reviewer 2 (Kevin Konrad)

Comment: “This manuscript provides new paleomagnetic data for an ancient Hawaiian plume derived seamount. The manuscript speaks to the debate over the origin of the Hawaiian-Emperor Bend (HEB) as well as the role of large low shear-wave velocity provinces play in modulating plume motion. The manuscript is well written with a clear and concise story. This manuscript is suitable for publication in Nature Communication with some minor revisions detailed below.”

Response: We thank the reviewer for this assessment.

Comment: “Using changes in inter-hotspot distance for absolute plume motion: If you wish to state absolute plume motion derived from comparing the geometry of Louisville to Hawaii, the authors need to clearly state that there is no eastward drift of the Louisville hotspot assumed. This eastward drift is predicted by Steinberger (2000) plume motion models as well as a slightly eastward, but mostly southward, drift from Hassan et al. (2016). If you model some eastward drift affecting the Louisville plume you will get slightly different values, although it is likely within uncertainty of your estimates.”

Response: We have tried throughout our manuscript to separate data from models. Therefore, our inter-hotspot calculations have been conducted in this context, without any numerical model assumptions. However, we appreciate the point raised by the reviewer. It is indeed worth noting that if there had been a significant component of eastward drift of the Louisville hotspot, this would increase distances between coeval seamounts on the Hawaiian-Emperor and Louisville Seamount chains. This is a function of the geometry of the Louisville track. This hypothetical eastward motion would thus act opposite to the observed decrease in distances between 63 and 52 Ma. Thus, if eastward motion occurred, our calculations might underestimate the actual separation change. We have added a note on this issue in the Supplement. Specifically, we state:

“Finally, we note that some models infer an eastward component of Louisville hotspot drift¹⁵. In our approach we explicitly attempt to separate observations from models. However, if the Louisville hotspot drifted eastward between 63 and 52 Ma, our calculations *underestimate* the rates of convergence and Hawaiian hotspot motion.”

We cite the Koppers et al. (2004) work that discusses eastward motion inferred from Steinberger’s models:

15. Koppers, A.A.P., Duncan R.A. & Steinberger B., Implications of a nonlinear $^{40}\text{Ar}/^{39}\text{Ar}$ age progression along the Louisville seamount trail for models of fixed and moving hot spots. *Geochem. Geophys. Geosyst.* **5**, Q06L02 (2004).

Comment: “Chemistry of the Emperor Chain Seamounts: There are alternative views to explain the chemical composition of the Emperor Seamounts that do not invoke changes plume location relative to the LLSVP. Regelous et al. (2003) interpreted the chemical systematics of the Emperor Chain seamounts as resulting

from changes in the thickness of the lithosphere the Hawaiian plume was interacting with. As the hotspot moved further way from the thinner lithosphere formed from the ancient Kula/Pacific Ridge, the degree of melting decreased resulting in more heterogeneous enriched magmas similar to what is observed on younger Hawaiian volcanos. This interpretation of the chemistry more closely supports a ridge capture/release scenario, although there is no chemical evidence that the plume was on the ridge prior to ~ 60 Ma. Work by Jones et al. (2017) argued that the Kea-type magmas (similar to the Emperor Chain) are formed from a difference in melting regime of the plume. To briefly summarize, the kea type magmas originate from hotter, deeper peridotite while the Loa lava flows form from shallower more-enriched pyroxenites. The geochemical trends result from changes in plate motion relative to the shearing of the mantle plume. Neither of these scenarios invoke changes in deep mantle dynamics.”

Response: We appreciate the work of Regelous et al. (2003), and the prior work of Keller et al. (2000), on early ridge interaction, and this is of course a consideration for any geodynamic explanation of the oldest extant Emperor chain (cf. discussion in Tarduno et al., 2003; 2009). The main issue we have focused on here is the temporal signature of the Kea and Loa trends. We further appreciate the alternative explanations posited for double geochemical trends on hotspot tracks; the work of Jones et al. (2016) was already referenced in our manuscript. But while the mantle flow/plate motion change mechanism proposed is consistent with geochemical observations from the Hawaiian islands, they are inconsistent with data from the Northwest Hawaiian Ridge, requiring more complex scenarios. However, we agree with the reviewer that we should perhaps be more explicit in alerting readers to alternative mechanisms proposed to explain double geochemical trends. We revised the relevant sentences as detailed below and added a second reference to the work of Jones et al.:

“...Emperor trend seamounts lack the Loa trend signature. Although there are several ways to explain double geochemical traces on young volcanic chains⁵², Harrison et al.⁵³ have focused on a mechanism that can give rise to the observed longterm temporal signature without a change in plate motion. Southward motion and anchoring of the plume on the LLSVP (Fig. 4A) results in the gradual entrainment of LLSVP material and the appearance of Loa lavas on the Hawaiian chain⁵³. This explanation is viable if...”

The citation added is as follows:

52. Jones, T.D. *et al.* The concurrent emergence and causes of double volcanic hotspot tracks on the Pacific plate. *Nature* **545**, 472-476 (2017).

Comment: “Plume attraction to LLSVPs: I advise caution with attributing plume motion to plume migration towards an LLSVP edge. I have not seen any models that predict this phenomenon (although I may have missed some). Harrison et al. (2017) suggests that method but doesn’t provide a geodynamic simulation to support it. Furthermore, Harrison et al. (2017) suggest the LLSVPs contain recycled subducted material, which is currently heavily debated (e.g. Li and McNamara, 2013; Davies et al. 2015; Rizo et al., 2016). However, many geodynamic models predict the initiation and anchoring of plumes on LLSVP margins (e.g. Steinberger et al. 2012; Tan et al. 2011). Therefore, I feel either more references are needed or more information supporting the claims in the abstract and text that LLSVP’s “...wander as they continually attract plumes...”.”

Response: The motion of plumes toward a dominant buoyant upwelling is a basic consequence required by lateral inflow to fulfill mass-conservation. This is discussed in a number of publications, but we note that these may be easily missed because to date much of the literature on mantle flow and LLSVP is simplified to the point that basic physical parameters cannot be evaluated, and/or pose sharp contradictions with what is known about the mantle. For example, in the quoted Tan et al. (2011) work, the thermal field is plotted rather than the velocity field, and the modeled temperatures in the mid-mantle yield temperatures and associated buoyancy that is incompatible with seismic data. However, we wholeheartedly agree that additional references and minor new text would clarify this situation.

First, regarding the interaction of plumes and large mantle upwelling, we can consider the following. Forte et al. (2010) (their Figure 5 and 6) presents the deep mantle velocity field predicted from global dynamic simulations, showing flow directed towards LLSVPs. This addresses the African LLSVP and is appropriate in light of our argument of LLSVP motion in the African hemisphere for the past 100 million years. Another relevant work is by Davies and Davies (2009). These authors consider high Rayleigh number 3D spherical convection models to show plume motions in the mantle wind (their Figure 1 and 2). Second, we can address whether LLSVPs are areas of large mantle upwellings. To answer this we refer to works that emphasize the thermal nature of LLSVPs. For example, recent work by Liu and Grand (2018), and prior work by Schuberth et al. (2012), nicely make the case for a mostly thermal (and thus upwelling) nature of LLSVPs. Finally, we also consider citation to work that points to the inevitability of LLSVP motion. A paper in Gondwana Research by Zhong and Liu (2016) makes the relevant point: it is dynamically infeasible to have stable LLSVPs and evolving plate configurations at the same time.

We combine these into the following additional text sentences, after our prior text “..This motion has implications for the nature and origin of LLSVPs....”. Specifically:

“Although their base (~ 100 km) may be chemically distinct, seismic evidence^{60–62} is consistent with LLSVPs being dominated by thermal heterogeneity. Global geodynamic flow models predict that the associated upwelling flow induces horizontal motion in the deep mantle directed towards the provinces⁶³. This deep mantle lateral motion helps to explain the migration of plumes reported in high Rayleigh number 3-D spherical mantle convection models⁶⁴.”

And in a revised final paragraph, we note:

“...it is dynamically infeasible to have stable LLSVPs and evolving plate configurations at the same time⁶⁷.”

The following references have been added:

60. Liu, C. & Grand, S.P. Seismic attenuation in the African LLSVP estimated from PcS phases. *Earth Planet. Sci. Lett.* **489**, 8-16 (2018).

61. Schuberth, B.S.A., Zoroli, C. & Nolet, G. Synthetic seismograms for a synthetic Earth: long-period P- and S-wave traveltime variations can be explained by temperature alone. *Geophys. J. Int.* **188**, 1393-1412 (2012).

63. Forte, A.M. *et al.*, Joint seismic-geodynamic-mineral physical modelling of African geodynamics: A reconciliation of deep-mantle convection with surface geophysical constraints. *Earth Planet. Sci. Lett.* **295**, 329-341 (2010).

64. Davies, D.R. & Davies, J.H. Thermally-driven mantle plumes reconcile multiple hot-spot observations. *Earth Planet. Sci. Lett.* **278**, 50-54 (2009).

67. Zhong, S. & Liu, X. The long-wavelength mantle structure and dynamics and implications for large-scale tectonics and volcanism in the Phanerozoic. *Gondwana Res.* **29**, 83-104 (2016).

Comment: “Line by line points:

Abstract: missing ‘shear’ in ‘large low velocity province (LLSVP)’”

Response: We thank the reviewer for catching this- it has been corrected.

Comment: “Figure S6: What is the hotspot location for Louisville? The current hotspot location is debated, assuming you’re using the data from Konrad et al. (2018) it’s based on the location suggested in Koppers et al. (2011).”

Response: For consistency with the age information, we indeed followed Konrad et al. (2018) which in turn follows Koppers et al. (2011). This choice does not affect our calculations. Nevertheless, we agree that the source for the youngest location plotted in Figure S6 should be documented and we have done so in the revised, associated caption, adding the following sentence:

“Although it does not factor in our distance calculations, we note that the youngest Louisville location shown is 1.1 Ma from Koppers et al.²⁷.”

The Koppers et al. (2011) citation has been added to the reference list:

27. Koppers, A.A.P. *et al.* New $^{40}\text{Ar}/^{39}\text{Ar}$ age progression for the Louisville hot spot trail and implications for inter-hot spot motion. *Geochem. Geophys. Geosyst.* **12** Q0AM02 (2011).

Reviewer 3 (Dietmar Müller)

Comment: “This is an excellent paper which was a joy to read. It presents important new data to clarify the cause of the Hawaii-Emperor Bend (HEB), a major ongoing controversy. The paper is clearly written and well-illustrated, and follows clear, deductive scientific reasoning, unlike some other papers that have been published on this subject. This paper forms an important milestone to cement our understanding of the HEB origin. I recommend that this paper be published essentially as is, with some very minor corrections.”

Response: We thank the reviewer for this assessment.

Comment: “Under the heading “Creating the HEB Morphology” I believe that an incorrect reference is given: The confirmation of rapid hotspot motion indicates that this process must be central to the formation of the HEB. This conclusion is in concert with the lack of clear geological evidence for a profound event, on the Pacific margins³⁹ or in the seafloor spreading record⁴⁰, Ref 40 is the reply to a comment on Wright et al.’s Geology paper from 2016 (ref 43). Ref 43 should be cited here instead of ref 40.”

Response: We thank the reviewer for noting this. Both references are of prime importance. The Wright et al. Geology paper discusses the lack of seafloor evidence for a change in plate motion commensurate with causation of the HEB, whereas the Reply nicely documents the plethora of other known Pacific plate motion changes that did not create hotspot track bends. We inadvertently duplicated citation to the Reply, thus omitting the original paper. This has been corrected.

Comment: “In the same section, there is this statement: Similar conclusions can be reached based on the alternative plate circuit, through Lord Howe Rise⁵, updated by the revised Antarctic-Australian spreading history⁴⁴ (fig. S9C). Even though I agree with the general statement, I should warn the authors that they should not be using the rotation from ref 44 (Whittaker et al., Science, 2007). Unfortunately there were some oversights in this paper, and the rotations we derived at the time did not work entirely well with fitting the westernmost portion of the SEIND spreading system between the Kerguelen Plateau and Broken Ridge. Therefore we later revised these rotations and designed them such that the Kerguelen Plateau-Broken Ridge region would be fitted better. The reference is Whittaker, J.M., Williams, S.E. and Müller, R.D., 2013, Revised Tectonic Evolution of the Eastern Indian Ocean, *Geochemistry, Geophysics, Geosystems*, 14, 6, 1891-1909. These rotations are also used in our current global plate model Müller, R.D., Seton, M., Zahirovic, S., Williams, S.E., Matthews, K.J., Wright, N.M., Shephard, G.E., Maloney, K.Y., Barnett-Moore, N., Hosseinpour, M., Bower, D.J. and Cannon, J., 2016, Ocean basin evolution and global-scale plate reorganization events since Pangea breakup, *Annual Review of Earth and Planetary Science*, Vol 44, 107-138. DOI: 10.1146/annurev-earth-060115-012211. And the associated rotation file be downloaded here: <https://www.earthbyte.org/category/resources/data-models/global-regional-plate-motion-models/> under “Müller et al. (2016) Late Triassic to present-day global model (*Annual Review of Earth and Planetary Sciences*)” I recommend that the authors redo their associated calculations with these rotations, as otherwise any lines of arguments that depend on these rotations could easily be shot down because the 2007 rotations are known to be flawed (even though the basic arguments in the paper still hold) and slight changes in these rotations can have very non-linear effects on far-away places on the Pacific Plate.

Response: We have adopted the new rotations as suggested. These do not affect our interpretations but are needed for completeness, as emphasized by the reviewer. We have replaced the Whittaker et al. (2007) citation with the Whittaker et al. (2013) citation and we have added the Müller et al. (2016) reference (see revised caption, Figure 1).

Comment: “In the second-last paragraph of the paper: LLSVPs should not be thought of as constant fixed features. Instead, they can exhibit a net slow motion punctuated by episodes of rapid wander I would recommend to modify this statement in the following sense. One of the points of the Hassan et al. (2016) paper was to show that rather than wholesale motion of LLSVPs, they appear to undergo deformation, and their sides or flanks that are most proximal to active subduction zones are much more prone to deformation and migration than sides far away from active subduction. Wholesale motion of LLSVPs might

have occurred as well in addition, and we see this most clearly with the example of the Perm anomaly, a small LLSVP which formed in response to a network of subduction zones which completely surrounded it until the Mongol-Okhotsk Ocean closed. The Perm anomaly is not in the place now where reconstructions (actually anyone's reconstruction) place the girdle of subduction zones that likely caused it, so it must have experienced wholesale lateral motion (Flament et al., 2017). This is an inescapable conclusion, unless one wanted to argue that the Perm anomaly has nothing to do at all with Mesozoic subduction history, and just happens to be in a nearby place to this now extinct subduction girdle by pure coincidence. So there is evidence for both (LLSVP asymmetric deformation and wholesale motion) and both processes may cause hotspot wander. It might be worthwhile making this point.”

Response: We thank the reviewer for noting this and urging us to better clarify the duality of apparent LLSVP motion due to deformation and whole scale motion. We have added a sentence that emphasizes that the large scale hotspot motion observed implies wholesale LLSVP motion, and we now cite the Flament et al. work on LLSVP motion in the 3rd-to-last paragraph of the manuscript.

Citation added:

59. Flament, N. Williams, S., Müller, R.D., Gurnis, M., Bower, D.J., Origin and evolution of the deep thermochemical structure beneath Eurasia. *Nature Comm.* **8**, 14164 (2017).

Comment: “Last paragraph:

A long history of subduction has likely focused lower mantle upwellings beneath Africa and the Pacific. I find this sentence ambiguous. What is a “long history”? Is this meant to be 100 or 500 million years? Even though I know what the authors mean to imply, it may not be so clear to a general readership. I think this argument should be made much more strongly and clearly.”

Response: We have clarified this in the revised manuscript as “post-Pangea.”

Comment: “Last paragraph:

Many published mantle convection models run over ~200 my produce the current overall LLSVP structure. Amongst all the models we have run in our group over more than a decade (easily more than 100, and many are published) one would not find a single one that did not result in this outcome. So one has to ask oneself: 200 million years of post-Pangea convection provides the overall geometry of the LLSVPs as they are observed, and, as recently pointed out in a paper on Stoneley wave splitting by Koelemeijer et al. (Nature Comms, 2017), they likely have an overall density that is lower than the surrounding mantle, not higher, as often assumed. This is in line with convection models that produce LLSVPs primarily as thermal upwellings, even though their base is likely chemically distinct - see recent EOS piece on this topic. So, simple deductive reasoning then tells us that if subduction history has such a strong influence on shaping LLSVPs, indeed creating them as they exist today, then obviously they cannot be fixed through time, as subduction geometry is not fixed in time. The point should to be made more strongly towards the end of the paper.”

Response: We thank the reviewer for this suggestion. The Koelemeijer et al. reference has been added (ref. 62). We have split the prior last paragraph into 2 paragraphs and added text that addresses these issues.

REVIEWERS' COMMENTS:

Reviewer #2 (Remarks to the Author):

All revisions were properly handled and I have no further comments.

Thank you for your thorough responses,

Kevin Konrad

Reviewer #3 (Remarks to the Author):

The authors have accommodated all reviewer comments, and I recommend that the paper be published as is.

Response to Reviewer Comments

As detailed below, the reviewers ask for no further revisions.

Reviewer 2 (Kevin Konrad)

Comment: “All revisions were properly handled and I have no further comments. Thank you for your thorough responses,”

Response: We thank the reviewer for this assessment.

Reviewer 3:

Comment: “The authors have accommodated all reviewer comments, and I recommend that the paper be published as is.

Response: We thank the reviewer for this assessment.